# Identification of Uric Acid Crystals Accumulation in Human and Animal Tissues Using Combined Morphological and Raman Spectroscopy Analysis

**DOI:** 10.3390/diagnostics12112762

**Published:** 2022-11-11

**Authors:** Teodoru Soare, Ana Maria Iordache, George Nicolae, Stefan-Marian Iordache, Cosmin Baciu, Silviu Marinescu, Raluca Ioana Rizac, Manuella Militaru

**Affiliations:** 1Department of Pathology, Faculty of Veterinary Medicine, University of Agronomic Sciences and Veterinary Medicine of Bucharest, Splaiul Independentei Street, No. 105, Sector 5, 050097 Bucharest, Romania; 2Optospintronics Department, National Institute for Research and Development for Optoelectronics—INOE 2000, 077125 Magurele, Romania; 3Department 14 Orthopedy-Traumatology-ATI, University of Medicine and Pharmacy Carol Davilla (UMFCD), Dionisie Lupu Street, No. 37, Sector 2, 020021 Bucharest, Romania; 4Clinical Emergency Hospital (SCUB) Floreasca Route, No. 8, Sector 1, 014461 Bucharest, Romania; 5Department 11-Plastic and Reconstructive Surgery, Pediatric Surgery, University of Medicine and Pharmacy Carol Davilla (UMFCD), Eroii Sanitari Bvd., No. 8, Sector 5, 050471 Bucharest, Romania; 6Discipline of Plastic Surgery and Reconstructive Microsurgery, Emergency Clinical Hospital “Bagdasar-Arseni”, Berceni Street, No. 12, Sector 4, 041915 Bucharest, Romania

**Keywords:** crystals, uric acid, Raman spectroscopy, gout, reptiles, human, comparative pathology, poultry

## Abstract

Gout is a metabolic condition, common to animals and humans, issuing from the excessive accumulation of end products of proteins degradation. In this study, histopathological and cytological examinations, combined with Raman spectroscopy, have been performed to investigate tissue samples from reptiles, chickens, and humans, presenting lesions produced by uric acid accumulation. As a result of classic processing and staining techniques commonly used in the anatomopathological diagnosis, uric acid crystals lose their structural characteristics, thus making difficult a precise diagnostic. Therefore, complementary diagnostic methods, such as Raman spectroscopy, are needed. This study compares from several perspectives the above mentioned diagnostic methods, concluding that Raman spectroscopy provides highlights in the diagnosis of gout in humans and animals, also adding useful information to differential diagnosis of lesions.

## 1. Introduction

Gout is a metabolic condition characterized by either excessive production, or insufficient excretion, of uric acid. The effect is an increase of the amount of uric acid in the blood, and consequently, the deposition of monosodium urate birefringent crystals in certain tissues.

In vertebrates, uric acid is derived from nucleic acids with exogenous and endogenous sources. They are degraded in the liver to release purine and pyrimidine bases. If these bases are not reused by the body, they will be further degraded and eventually excreted. Pyrimidine bases are catabolized to carbon dioxide and ammonia [1]. In certain vertebrates, including lizards and some turtles, the breakdown of purine bases can produce very large amounts of uric acid. Degradation of adenine and guanine—the main purine bases—turns into xanthine, and through the xanthine oxidase, uric acid is formed [2].

Uric acid is present in the blood predominantly as monosodium urate. Both uric acid and urate salts, are relatively insoluble in water. When their concentrations increase, either in the blood (hyperuricemia), or in other fluids (synovial fluid), monosodium urate crystallizes and forms insoluble precipitates that are deposited in various organs and tissues. Thus, depending on the site where monosodium urate will crystallize, visceral gout, articular gout, and periarticular gout, are distinguished [3].

From the point of view of etiology, the literature most frequently cites nutrition as the main predisposing factor in animals and humans [4]. In addition to food factors, there are situations, in which gout appears on a toxic background, e.g., in the administration of certain drugs. Another main cause of gout in animals is represented by husbandry problems linked to humidity of the environment, or the ease of access or quality of water [5,6].

Identifying the aspects of gout in humans and animals requires the corroboration of clinical and paraclinical investigations. In reptiles and birds, the clinical signs differ depending on the location of the disease. Thus, articular and periarticular gout produces painful swelling of the joints, manifested by lameness, reluctance or refusal to move. Visceral gout, on the other hand, is difficult to diagnose clinically, as it may not present clinical signs or, if present, may be nonspecific, such as anorexia or lethargy [7,8]. Blood biochemistry is not a foolproof tool [9,10]. The radiographic examination can reveal lytic lesions at the articular or periarticular level. Gout tophi only if impregnated with calcium salts may become visible on X-ray. Regarding the kidneys, nephromegaly and enhanced from normal radio-opacity can be observed radiographically. Uric acid deposits impregnated with calcium salts can be distinguished as hyperechoic structures by ultrasound [11,12]. In humans, the clinical signs, the paraclinical investigations, and their interpretations are similar to the ones mentioned in animals. Additionally, humans describe the pain associated with joint gout as a burning sensation.

It is known that crystalline material accumulated within different tissues can cause granulomatous inflammatory reactions, foreign body type, in most of the studied species [13,14]. However, these crystal structures are lost by dissociation following the usual histoprocessing techniques, especially due to the formalin, which is the most commonly used fixative [15,16].

Almost 100 years ago, Sir Chandrasekhara Venkata Raman, assisted by his student Kariamanikkam Srinivasa Krishnan, discovered the phenomenon of inelastic scattering of light, also known as the Raman effect [17], which was predicted theoretically by Adolf Gustav Stephan Smekal in 1923 [18].

Raman spectroscopy is a nondestructive and effective tool that has been applied successfully to various fields such as: medicine [19,20,21,22,23], dental research [24,25,26], biology [27], food safety [28], chemistry [29], pharmacy [30], etc. Material identification and its structural characterization are complex, since Raman scattering acts at various levels, from surface to the depth of the samples. It is sensitive to chemical effects, such as crystallinity, polymorphism, phase discrimination, hydrogen bonding. Two materials expressed through the same chemical formula would display different Raman spectra due to the different organizations of the bonds building the respective molecules. High spectral resolution is necessary when analyses of samples suspected of polymorphism are required. A quick reading of a Raman spectrum will rely on peaks positions (profile of molecular structure, phase, and stress/strain), peaks heights (information on material concentration and distribution), and peaks width (knowledge of crystallinity and phase). Since Raman spectroscopy is based on vibration of molecules when irradiated with a specific wavelength and thus “fingerprinting” each molecule, it can be applied in the field of veterinary and human medicine to sustain the histopathological diagnosis. A major advantage of the Raman spectroscopy is that it can be used both in vivo and in vitro to assess different diseases. This provides real-time monitoring of the treatment and intervention, which limits the “guesswork” in many clinical fields [31]. Oncology is one of those fields that could take advantage of the rapid investigation performed on surgical margins by Raman spectroscopy in order to assess the malignancy/nonmalignancy of the excised/left tissue [32].

However, there are few papers connecting the veterinary and human medicine. Here, we conduct an exploratory investigation to use Raman spectroscopy in both branches of medicine. Therefore, we found necessary to use Raman spectroscopy as a complementary investigation technique to sustain a definitive diagnosis. This study is a descriptive investigation, which aims to analyze and compare the benefits of optical microscopy techniques and Raman spectroscopy, and suggests the latest as a more suitable approach, to diagnose gout in animals and humans.

## 2. Materials and Methods

The study was conducted on four reptiles (*Python regius*, *Python bivittatus*, *Pantheroidus guttatus*, *Chamaeleo calyptratus*), one poultry (*Gallus gallus domesticus*), and three human samples.

The reptiles and the poultry were submitted for necropsy examination to the Pathology Department of the Faculty of Veterinary Medicine, University of Agronomic Sciences and Veterinary Medicine of Bucharest. During the gross examination, samples were collected for histopathology and cytology. Hematoxylin and Eosin (HE), modified Von Kossa, and Gomori histopathological stains have been used. All slides were examined using both light microscopy and polarized light microscopy, by means of an Olympus BSH microscope. Photos were taken using an Olympus SC50 microscope camera and Olympus CellSens software.

The human samples were provided by the Bucharest Clinical Emergency Hospital. They resulted from orthopedic surgeries and have been used in this study with the approval of the patients and underwent microscopical examination as the aforementioned animal samples.

Fresh samples, twins of the above mentioned ones, and slides, were sent to the National Institute for Research and Development for Optoelectronics for performing Raman spectroscopy. The Raman spectra were acquired using a LABRAM HR 800 (Horiba JobinYvon) micro-Raman spectrometer, in the backscattering geometry with a λ = 633 nm HeNe laser as excitation source giving a power of 2–2.2 mW on the sample. The Raman wavenumber shift measured by the system was calibrated using the known 520.7 cm^−1^ peak of a Si wafer. An 1800 gr/mm diffraction grating was used and the spectral range extended from 200 to 4000 cm^−1^ to cover all vibrations of biological interest, i.e., S-S, C-S, C-C, C-N, C-O, OH, NH, and H_2_O. The recorded Raman spectra were cut at 2500 cm^−1^ as the large wavenumber region did not contain significant spectral information. Measurements were taken using a 50X objective and the final spectrum was obtained as the average of five consecutive spectra, each collected for 10 s. Raman data were exported from the Raman instruments operating software and analyzed using OriginPro v2017 (OriginLab Corporation, MA, USA). Spectra were baseline corrected, normalized, and smoothed using a Savitzky-Golay smoothing. Spectrum complexity is caused by overlapping peaks, and to expose constituent peaks, deconvolution across large spectral regions is required. A Gaussian profile was used for peak fitting

Analytical grade reagents were used without additional purification. Uric acid (≥99%, crystalline) was purchased from Sigma Aldrich, Hematoxylin & Eosin (HE) Staining Kit from VECTOR Laboratories, Newark, NJ, USA, CVK-1-IFU Calcium Stain Kit (Modified Von Kossa) from Scytek Laboratories INC, Logan, Utah, USA, and Gömöri trichrome stain from Remed Prodimpex SRL, Pantelimon, Romania.

## 3. Results

This study was focused on animal and human specimens that exhibited macroscopic lesions characteristic to gout. All the investigations relied on the qualitative assessment of the uric acid present in samples. The study used several methods to confirm the diagnosis and to rule out other similar conditions. Raman spectroscopy was then used as a new method of diagnosis and the specificity of this technique was assessed for native smears and histopathological slides.

(a)Macroscopic analysis

From a macroscopic point of view, gout in reptiles is characterized by yellowish-white lines, or spots that can be observed in the structure of the renal parenchyma (Figure 1A) [5,6]. During a mild stage of the condition, the kidney has an increased volume (degenerative aspect), and is pale. Based on the species, some reptiles might exhibit organ or joint involvement linked to the articular and periarticular presence of uric acid crystals (Figure 1B). The avian gout gross lesions are similar to the ones described in reptiles (Figure 1C). The renal gout is recognized by the presence of multifocal white, chalky, sandy deposits, in both subcapsular and on cross section, and is usually associated with marked dilatation of ureters due to uric acid accumulation. In humans, the main lesions associated with gout involve the interphalangean joints with possible extension to larger joints (elbow) and exhibit severely increased thickness, and evident distention of the articular capsule, along with marked white, chalky, deposits of uric acid (Figure 1D).

In gout samples, the specific histoprocessing using anhydrous techniques is the key to preserve the uric acid crystals. If the uric acid deposits exceed the basement membrane of the renal tubules, gout tophi form (Figure 2A). They appear as radiating acicular deposits of uric acid crystals surrounded by an inflammatory reaction. In acute cases, degenerated heterophils are observed through histopathology (Figure 2B), and as the lesion becomes chronic, macrophages, lymphocytes and multinucleated giant cells appear. Gout tophi are most commonly located in the renal tubules, but they can also be found in the interstitium or in the glomerular mesangium. In severe cases, these structures may coalesce, giving rise to macroscopically visible whitish foci [5,6]. In humans, the histopathological aspect is characterized by a granulomatous foreign body type reaction (Figure 2C). The multinucleated giant cells surround a central area of necrosis with the presence of acicular crystals.

The routine histopathology, unfortunately, will result in the dissociation of the uric acid crystals [15,33]. Therefore, the pathologist will observe only the inflammatory reaction, composed of foreign body giant cells, macrophages, heterophils in reptiles and birds, surrounding areas with loss of morphological details (necrosis) and colorless radially arranged optically empty spaces mimicking the crystals (Figure 2A,D).

The main differential diagnosis in articular gout is made with pseudogout. The latest is produced by the accumulation of calcium salt within the joints. Clinically and grossly, this lesion is similar to gout. In this study, the Von Kossa stain was a helpful method to infirm the presence of calcium deposits (Figure 2E–G). The Gomori stain was used to emphasize the presence of uric acid crystals. Since the fixative for the samples used in this study was represented by formalin, the Gomori stain was negative in all examined sections, in all species (Figure 2H) [19].

In some of the cases cytological examination was also used. The native smears examined in light microscopy revealed an abundant, granular to acicular, amorphous, gray to black material (Figure 2I). Employing of polarized light examination, the gout crystals were bright and birefringent, suggestive of uric acid crystals accumulation (Figure 2J–L) [34].

(b)Raman analysis

First, the Raman signature of the uric acid was examined. Figure 3A shows a typical Raman spectrum of laboratory grade uric acid powder used as etalon, while (B) is a comparison between samples containing uric acid crystals as sustained by histopathology analyses. Raman bands located at 469, 559, 624, 791, 880, 995, 1036, 1119, 1285, 1403, 1497, 1592 and 1649 cm^−1^ are specific to uric acid [19,20,35,36,37]. Ring vibrations are assigned to the bands at 469, 500, 559, 624, 791, and 995 cm^−1^. The bands at 469 and 500 cm^−1^ are assigned to C-N-C vibrations in the ring. The bands in the 500–650 cm^−1^ region are the skeletal ring deformation [19,20,35,36]. The intense and sharp band at 624 cm^−1^ originates from the ring breathing mode [19,20,35]. The bands at 700–900 cm^−1^ are N-H out of plane, and in-plane, bending vibrations [36,37]. The 900–1300 cm^−1^ is a region of intense mixed vibrations modes, which include ring vibration, C-O, C-C, C-N stretching, O=H deformation and N-C-C stretching and bending vibration. Bands above 1500 cm^−1^ in the Raman spectra are assigned to the C-C (1497 cm^−1^) and C-N (1592 cm^−1^) stretching modes of nucleic acids [35,36]. The band at 1649 cm^−1^ can be assigned to the C=O stretching b and [36].

Raman analyses of the specimen (Figure 4, Figure 5, Figure 6, Figure 7 and Figure 8) followed the uric acid study. The spectral intensities of the samples were different from each other. Specific assignments of individual peaks could be found in Table 1. The authors consider that the biologically deposited uric acid crystals, as seen from histopathology analyses, have slightly different molecular bond energies, which result in the minute shifts in the Raman peak wave numbers reported here, as compared to the pure uric acid sample.

## 4. Discussion

This study emphasizes the weaknesses of histopathology examination in particular cases, where its sensitivity, and specificity, are drastically diminished mostly due to the reagents used in samples processing. An association with cytopathology is crucial in determining a diagnosis. The literature suggests that anhydrous fixatives (alcohol 95–100%) and stains [7,15,33] should be used for histopathology processing in order to highlight metabolites with crystalloid forms, accumulated or built up in different tissues, usually causing inflammatory type reactions [13,14] defined as foreign body reactions (granulomas).

Routinely, more than 12 h passes from the collection of the sample to its processing. Moreover, the regular processing of the sections involves the use of formalin as a fixative and the HE staining, thus causing degradation of the crystals, and hence the difficulty to recognize them by optical microscopy. These drawbacks are overcome using Raman spectroscopy, which can be used in situ to identify the molecular traces of the accumulated substances to obtain the specific spectra of the crystals [15].

Therefore, neither the histopathological, nor the cytopathological examination can provide a definite diagnosis on their own in particular cases. Another difficulty comes from the setup of the differential diagnosis in joint lesions such as gout, pseudogout, autoimmune inflammatory processes or of a different nature than the accumulation of metabolites, and, last but not least, neoplasms. Therefore, the need was to identify a diagnostic method that would solve all these problems. This study shows that Raman spectroscopy is a highly specific, very fast method, which does not require an elaborate processing of the samples, is cost effective, being comparable to cytology while looking at the subcellular level, and offers more information that can be used in the aforementioned differential diagnosis. Additionally, it can successfully support the other methods by detecting small amounts of uric acid from already processed samples, including in situations of uncertainty or second-opinion cases.

Histo-cytological examination can be performed easily on bulk samples, where the crystals are overgrown onto large particles. However, the configuration of the uric acid crystal is affected by many factors: temperature, protein vicinity and diseases (in the living body), ultraviolet light and organic solvent (when the sample is extracted and stabilized for diagnosis). Identification of uric acid crystals in bodily fluids extracted e.g., via arthrocentesis (surgical procedure that removes the excess fluid from the joints) could offer an insight into the metabolic chain of uric acid elimination and deposition.

Compared to previous studies dealing with the identification of uric acid via Raman spectroelectrochemistry [37] and surface-enhanced Raman scattering [38], our approach enabled faster and easier analysis of the specimens. Unfortunately, the limited availability of the specimens precluded a rigorous statistical analysis of the results.

Nevertheless, although histopathology remains the gold standard for specimen analysis, a faster and reliable method such as Raman spectroscopy may be used as a first line of diagnosis, supplementing cytological analysis and eliminating tissue removal.

## 5. Conclusions

This study suggests that Raman spectroscopy might be a suitable tool to analyze uric acid accumulation in tissue, as an indicator of potential gout presence. Unlike the other techniques, such as polarized light microscopy or simple sampling and ex vivo analysis with cytology, Raman spectroscopy is more specific, as it analyzes biochemical changes in the tissue at the subcellular level, while is also fast and can be used in situ. Following the results obtained through Raman spectroscopy, it can be remarked that this technique with high sensitivity and specificity can be used in the examination of gouty lesions in different species.

## Figures and Tables

**Figure 1 diagnostics-12-02762-f001:**
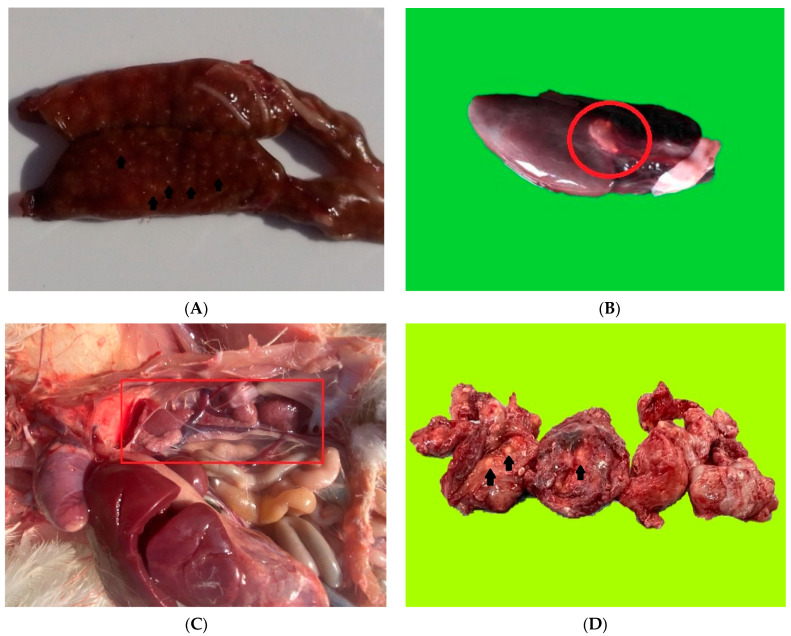
(**A**) Chameleon. White-yellow foci of uric acid within kidney parenchyma. (**B**) Snake. Heart—white chalky deposit (red circle). (**C**) Poultry. Coelomic cavity. Enlarged, pale, granulary-surfaced kidney (red box mark)—degenerative aspect. (**D**) Human. White chalky foci within hand joints.

**Figure 2 diagnostics-12-02762-f002:**
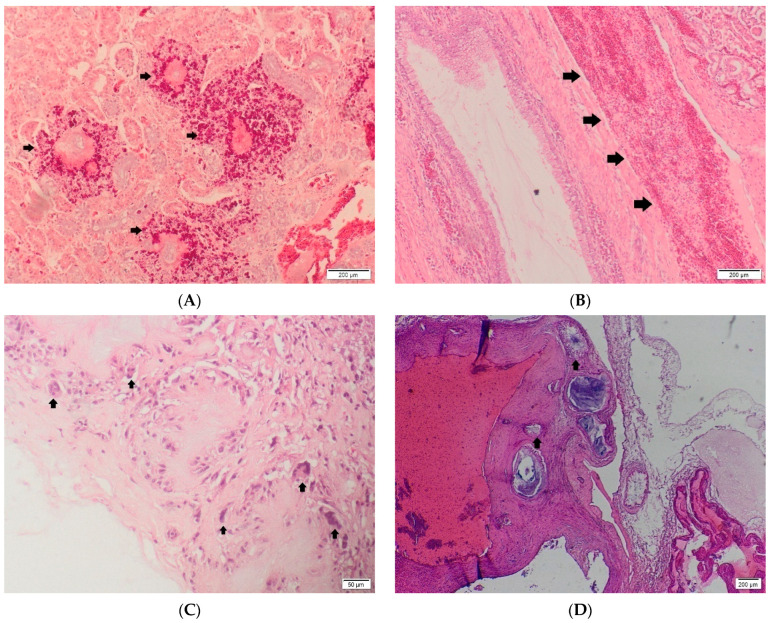
(**A**) Chameleon. Kidney. Gout tophi (black arrows) with severe granulomatous inflammatory reaction. HE, Ob. 4. (**B**) Poultry. Ureter cross section. Ureter ectasia and periuretral heterophillic infiltrate (black arrows). HE, Ob. 4. (**C**) Human. Severe granulomatous reaction within the joint capsule. Necrotic foci surrounded by multinucleated foreign body type cells (black arrows). Ob. 10. (**D**) Snake. Heart—cross section at the base of heart with several granulomatous foci involving the vascular wall (black arrows), HE Ob. 2. (**E**) Snake. Heart—Similar cross section–granulomatous foci (black arrows), Von Kossa Ob. 4. (**F**) Poultry. Ureter cross section. Diffuse negative content with discrete black Von Kossa positive material (black arrows). Von Kossa, Ob. 4. (**G**) Human. Joint capsule cross section. Von Kossa. Ob. 4. (**H**) Snake. Heart—Similar cross section–granulomatous foci (black arrows), Gomori Ob. 4. (**I**) Chameleon. Native smear from cloaca content. Different sized black crystals in light microscopy. Ob. 4. (**J**) Poultry. Native smear from ureter, polarized light, Ob. 10. (**K**) Chameleon. Native smear from ureter content, crystals in polarized light–(red arrows). Ob. 10. (**L**) Human. Native smear from joint content. Crystals in polarized light (yellow arrows). Ob. 4. C.

**Figure 3 diagnostics-12-02762-f003:**
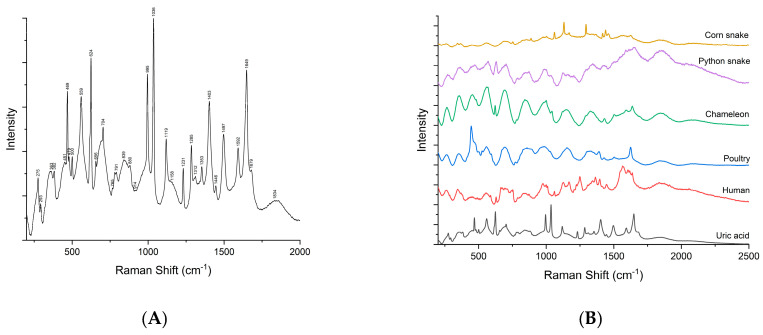
Raman spectra of uric acid (**A**) and comparison between samples vs. uric acid (**B**).

**Figure 4 diagnostics-12-02762-f004:**
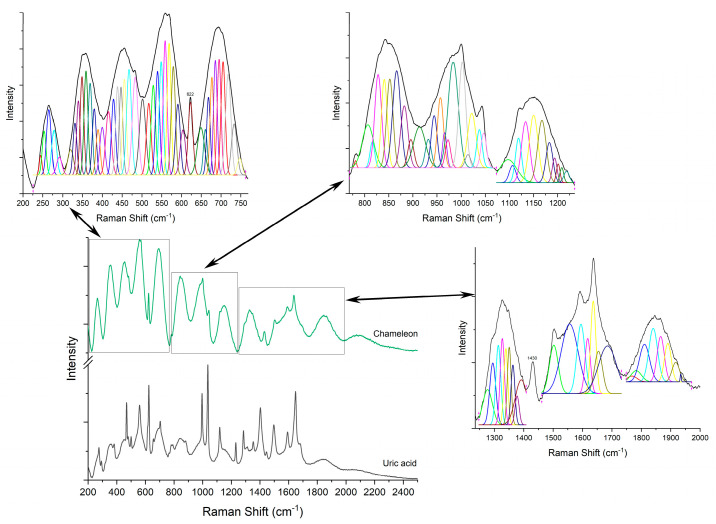
Raman spectra of pure uric acid and, respectively, chameleon sample.

**Figure 5 diagnostics-12-02762-f005:**
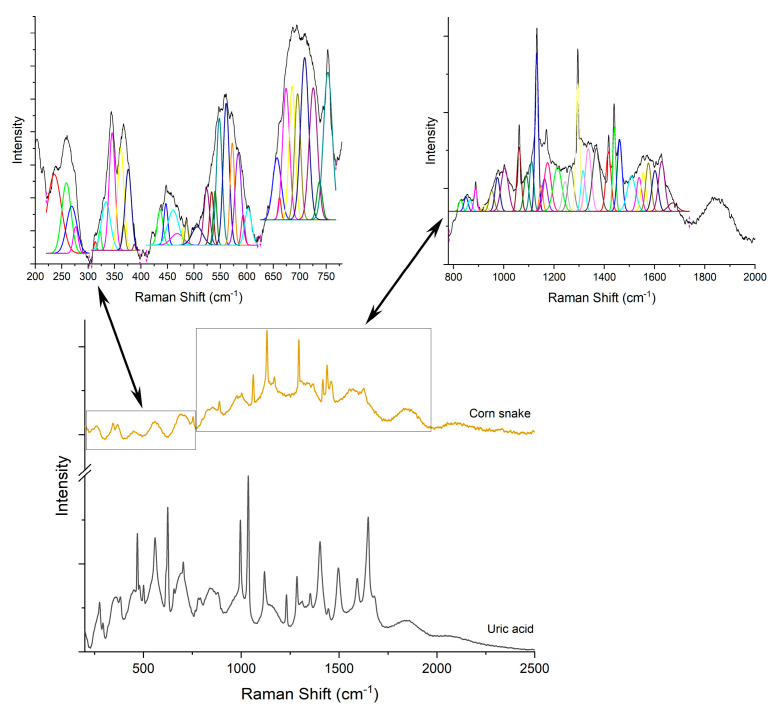
Raman spectra of pure uric acid and, respectively, corn snake sample.

**Figure 6 diagnostics-12-02762-f006:**
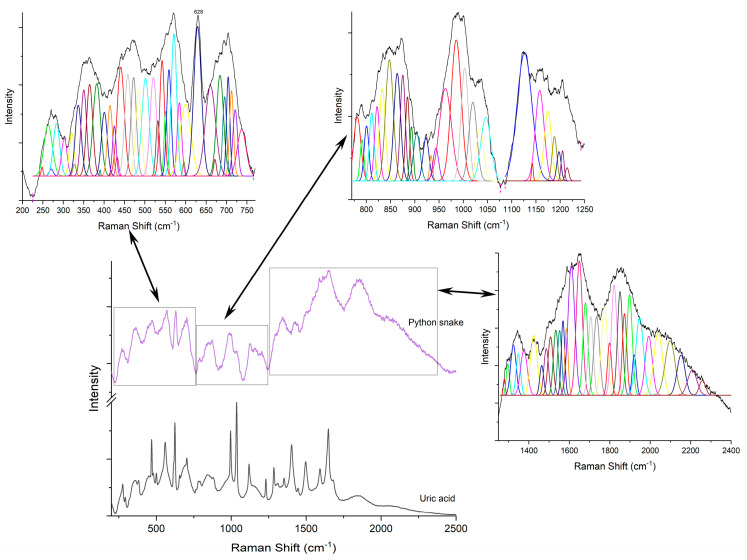
Raman spectra of pure uric acid and, respectively, Python snake sample.

**Figure 7 diagnostics-12-02762-f007:**
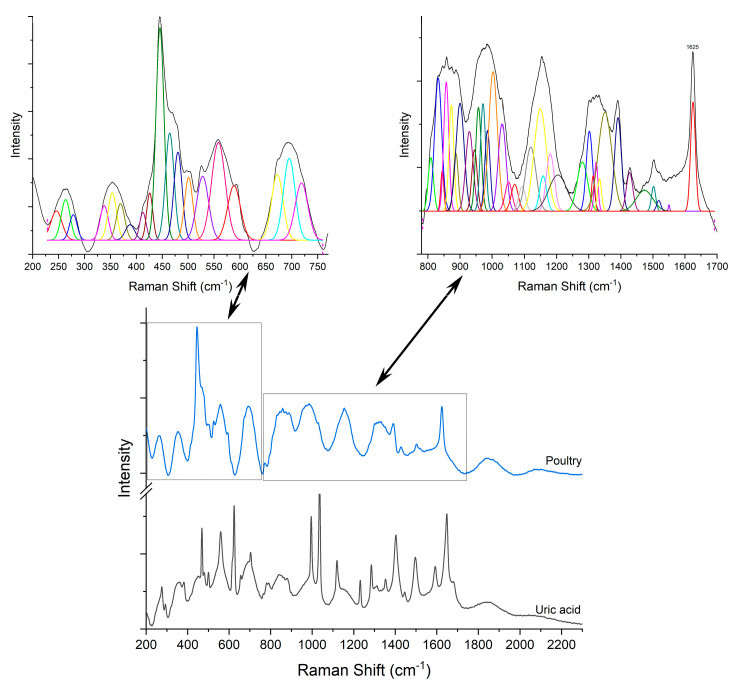
Raman spectra of pure uric acid and, respectively, poultry sample.

**Figure 8 diagnostics-12-02762-f008:**
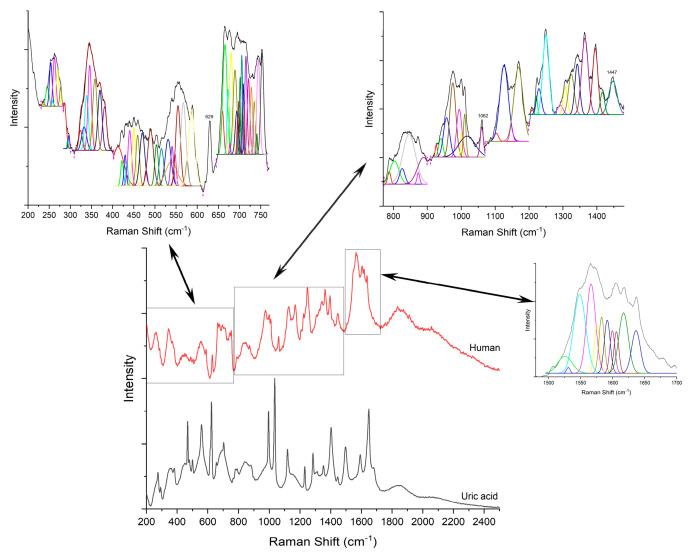
Raman spectra of pure uric acid and, respectively, human sample.

**Table 1 diagnostics-12-02762-t001:** Raman bands assignments of samples vs. uric acid.

Assignments		Raman Shift (cm^−1^)					
		Uric Acid	Chameleon	Corn Snake	Python Snake	Poultry	Human
C-N-C vibrations in the ring		469	467	468	472	464	470
		500	501	506	501	500	503
skeletal ring deformation	ring vibration	559	558	560	558	558	554
	ring breathing mode	624	622	-	628	628	629
N-H out of plane and in-plane bending vibrations		791	780	-	791	-	787
		880	882	887	884	886	888
ring vibration, C-O, C-C, C-N stretching, and N-C-C stretching and bending vibration	ring vibration	995	1002	1003	1003	1002	994
	C-O stretching	1036	1037	-	1034	1030	-
	C-N stretching	1119	1118	-	1126	1119	1126
	O=H deformation	1285	1276	1294	1295	1280	1291
C-N stretching		1403	1391	1416	1424	1391	1416
C-C stretching modes of nucleic acids		1497	1502	1507	1506	1501	-
C-N stretching modes of nucleic acids		1592	1594	1601	1585	-	1591
C=O stretching band		1649	1654	-	1649	-	-

## Data Availability

Not applicable.

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
