# Peer review of "Identification of Uric Acid Crystals Accumulation in Human and Animal Tissues Using Combined Morphological and Raman Spectroscopy Analysis"

_diagnostics, 2022, doi:10.3390/diagnostics12112762_

Round 1

Reviewer 1 Report

Manuscript Number: Diagnostics 1954568

Title: Morphological and Raman Spectroscopy identification of uric 2 acid crystals on different tissues in Comparative Pathology

Comments

The authors report Raman assay for uric acid for diagnosis of gout.  The work looks interesting, but it has got so many limitations that should be addressed before publication. I have the following comments.

1. Authors need to write some background information about Raman spectroscopy so that readers can capture what thy are talking. I recommend the following citation:

https://doi.org/10.1016/j.vibspec.2018.06.013

2. Lines 94-106, is not important in results section; better to take to the introduction section.

3. Figure 5 are not clear; needs to be redrawn. Is that solid Raman spectrum of uric acid? The figure caption is not self-explanatory.

4. Line 148, “Figure 5 (A) shows typical Raman spectra…” it is only one spectrum, please correct spectra to spectrum.

5. I can’t see the importance of enlarged bands taken from the Raman spectra of Fig. 6 -10.

6. Why don’t authors show the SERS spectra?

7. Lines 174-181, the paragraph is misleading. I would suggest the authors to consider this.

8. What is authors’ explanation to assign the bands only for uric acid? What about the bands are from other biological matrixes? Like proteins, amino acids , etc. In other words, how could the authors proof the selectivity of Raman to uric acid? We can observe many Raman band shifts. What is the explanation for this? What is their evidence to make a conclusion for assignment of bands based on the [solid] Raman spectrum of uric acid?

9. I can’t see the novelty of the work.

Author Response

Reviewer 1

The authors report Raman assay for uric acid for diagnosis of gout. The work looks interesting, but it has got so many limitations that should be addressed before publication. I have the following comments.

1. Authors need to write some background information about Raman spectroscopy so that readers can capture what thy are talking. I recommend the following citation: https://doi.org/10.1016/j.vibspec.2018.06.013

Authors introduced a paragraph related to Raman spectroscopy as the reviewer suggested.

2. Lines 94-106, is not important in results section; better to take to the introduction section.

Thank you for your valuable suggestion. The modifications were done according to reviewer suggestion. Authors moved the paragraph in the introduction section.

3. Figure 5 are not clear; needs to be redrawn. Is that solid Raman spectrum of uric acid? The figure caption is not self-explanatory.

All figures are at 600 dpi. Somehow Microsoft word processor cutoff from resolution. All figures are attached as individual files on the submission platform.

4. Line 148, “Figure 5 (A) shows typical Raman spectra…” it is only one spectrum, please correct spectra to spectrum.

The modifications were done according to reviewer suggestion.

5. I can’t see the importance of enlarged bands taken from the Raman spectra of Fig. 6 -10.

Spectrum complexity is caused by overlapping peaks, and to expose constituent peaks, deconvolution across large spectral regions is required.

6. Why don’t authors show the SERS spectra?

Authors did not perform SERS, just Raman.

7. Lines 174-181, the paragraph is misleading. I would suggest the authors to consider this.

The paragraph was reconsidered.

8. What is authors’ explanation to assign the bands only for uric acid? What about the bands are from other biological matrixes? Like proteins, amino acids , etc. In other words, how could the authors proof the selectivity of Raman to uric acid? We can observe many Raman band shifts. What is the explanation for this? What is their evidence to make a conclusion for assignment of bands based on the [solid] Raman spectrum of uric acid?

The authors focused on uric acid since its presence is a confirmation for gout diagnosis. Other biological matrix are beyond the scope of this paper.

9. I can’t see the novelty of the work.

The novelty of this work is given by the use of Raman spectroscopy for sustaining the histopatological analyses which confirm the diagnosis of gout.

Also, for the cases in which during histopathological processing the uric acid crystals will dissociate, Raman spectroscopy is a most accurate and specific method that can be used.

Reviewer 2 Report

1.      How did authors pick out the animals diagnose with gout? How to measure the degree of gout of reptile and chicken?

2.      The authors mention ‘spectral range extended from 200 to 4000 cm-1 to cover all vibrations of biological interest, S-S, C-S, C-C, C-N, C-O, OH, NH, and H2O’. However, spectral range of Raman spectra in manuscript only cover 200-2500cm-1. The expression needs to be modified.

3.      The laser intensity of Raman spectroscopy is not mentioned in Materials and Methods section.

4.      The data processing of Raman spectra in Figure 6-10 seems inappropriate. What model is used in peak extraction of Raman spectra. Multi-Gaussian? Lorentzian function? And curve-fitted Raman spectra need to be shown as comparation. The data processing methods should be described further in detail.

5.      The authors mention that ‘The Raman shifts at 469, 559 and 624 cm-1 representing signature peaks of uric acid crystals were present in all samples’. However, these Raman shifts alone do not indicate the presence of uric acid. It is important to provide Raman spectra of normal tissue to confirm the difference.

6.      The Y axis value is missing in all Raman spectra, causing the intensity of Raman peak to be unknown.

7.      The Raman spectra were collected from different part of tissue and average or from the same part?

Author Response

Reviewer 2

1. How did authors pick out the animals diagnose with gout? How to measure the degree of gout of reptile and chicken?

Thank you for your kind information and critical review. The reptiles and the poultry were submitted for necropsy examination to the Pathology Department of the Faculty of Veterinary Medicine, University of Agronomic Sciences and Veterinary Medicine of Bucharest. The human samples were provided by the Bucharest Clinical Emergency Hospital.

2. The authors mention ‘spectral range extended from 200 to 4000 cm to cover all vibrations of biological interest, S-S, C-S, C-C, C-N, C-O, OH, NH, and H O’. However, spectral range of Raman spectra in manuscript only cover 200-2500cm . The expression needs to be modified.

All collected spectra were acquired in the range 200-4000 cm-1, but the recorded Raman spectra were cut at 2500 cm-1 as the large wavenumber region did not contain significant spectral information.

3. The laser intensity of Raman spectroscopy is not mentioned in Materials and Methods section.

Laser power was added in text.

4. The data processing of Raman spectra in Figure 6-10 seems inappropriate. What model is used in peak extraction of Raman spectra. Multi-Gaussian? Lorentzian function? And curve-fitted Raman spectra need to be shown as comparation. The data processing methods should be described further in detail.

Spectrum complexity is caused by overlapping peaks, and to expose constituent peaks, deconvolution across large spectral regions is required. For peak fitting was used a Gaussian profile. The details for Raman processing are provided in Section 2. Materials and methods.

5. The authors mention that ‘The Raman shifts at 469, 559 and 624 cm representing signature peaks of uric acid crystals were present in all samples’. However, these Raman shifts alone do not indicate the presence of uric acid. It is important to provide Raman spectra of normal tissue to confirm the difference.

‘The Raman shifts at 469, 559 and 624 cm representing signature peaks of uric acid crystals were present in all samples’ was removed from the text.

6. The Y axis value is missing in all Raman spectra, causing the intensity of Raman peak to be unknown.

The Raman spectra is normalized (0-100). In our case intensity is not mandatory.

7. The Raman spectra were collected from different part of tissue and average or from the same part?

Raman spectra were collected from different areas of the tissue and averaged.

Reviewer 3 Report

This paper described an interesting study of developing new technology for uric acid crystal identification using morphological and Raman spectroscopy. The gold standard for establishing a definite diagnosis of gout is the presence of crystals in aspirated joint fluid or tophus. They have identified the common challenges of this standard gold technique, such as dissociation of the crystal structure during fixation, and then developed a method to sustain a definitive diagnosis. The study provides novel information and technology to advance the uric acid diagnosis field.

However, the following area needs to improve. 

In the introduction section:

Add a paragraph about why Raman spectroscopy is important in this particular study.

In the materials and methods section

1: In section lines 64-65, all species names should be italic according to the zoology nomenclature

2: line 71 HE stain needs to be expanded before abbreviating. (Hematoxylin & Eosin)

In the Results section:

Starting from paragraphs 2 to 6 is hard to follow as all the figure captions are everywhere. For example, paragraph 2 (starting line 107) discussion starts with figure 3A and then moves to Figures 1 A, 2A, and 4A. 

Further paragraphs also randomly discuss all the figures, making reading difficult for the reader. Therefore strongly suggest fixing that starting from Figures 1-4. Keeping each paragraph for A, B, C, and D images is okay. But need to follow the same flow. 

Line 109, the idea is repeated. 

Line 153 and 154 add citations.

In Table 1, why are some numbers in the bracket?

It would benefit the scientific community  If they could explain how they obtained sub-peaks by curve fitting, as shown in Figure 6-10 inlets.

 Also, have you conducted a Raman spectroscopy study for stain-fixed samples? If that data can be included, it will confirm the dissociation of crystals during fixation solvents. 

Please include sample preparation for Raman spectroscopy of specimens.

Author Response

Reviewer 3

This paper described an interesting study of developing new technology for uric acid crystal identification using morphological and Raman spectroscopy. The gold standard for establishing a definite diagnosis of gout is the presence of crystals in aspirated joint fluid or tophus. They have identified the common challenges of this standard gold technique, such as dissociation of the crystal structure during fixation, and then developed a method to sustain a definitive diagnosis. The study provides novel information and technology to advance the uric acid diagnosis field.

However, the following area needs to improve.

Thank you for your kind information and critical review.

In the introduction section:

Add a paragraph about why Raman spectroscopy is important in this particular study.

Thank you for your valuable suggestion. Authors added a paragraph related to the importance of Raman spectroscopy in this case.

In the materials and methods section

1: In section lines 64-65, all species names should be italic according to the zoology nomenclature
2: line 71 HE stain needs to be expanded before abbreviating. (Hematoxylin & Eosin)

The modifications were done according to reviewer suggestion.

In the Results section:

Starting from paragraphs 2 to 6 is hard to follow as all the figure captions are everywhere. For example, paragraph 2 (starting line 107) discussion starts with figure 3A and then moves to Figures 1 A, 2A, and 4A.
Further paragraphs also randomly discuss all the figures, making reading difficult for the reader. Therefore strongly suggest fixing that starting from Figures 1-4. Keeping each paragraph for A, B, C, and D images is okay. But need to follow the same flow.

Line 109, the idea is repeated.

Line 153 and 154 add citations.

In Table 1, why are some numbers in the bracket?
It would benefit the scientific community If they could explain how they obtained sub-peaks by curve fitting, as shown in Figure 6-10 inlets.

Also, have you conducted a Raman spectroscopy study for stain-fixed samples? If that data can be included, it will confirm the dissociation of crystals during fixation solvents.
Please include sample preparation for Raman spectroscopy of specimens.

Authors rearranged the paragraph according to reviewer suggestions.

Line 109 was rearranged.

Citations were added.

The brackets were removed.

Spectrum complexity is caused by overlapping peaks, and to expose constituent peaks, deconvolution across large spectral regions is required. For peak fitting was used a Gaussian profile.

Raman spectroscopy was done on fresh samples (human) and stained-fixed slides (animals).

Round 2

Reviewer 1 Report

Still comment number 8 is not addressed. I was asking how are you going to avoid the interferences from other biomolecules, but the authors respond about the focus of their research.